# Latent Speech-Text Transformer

**Yen-Ju Lu**[1◇]**, Yashesh Gaur**[2]**, Wei Zhou**[2◇]**, Benjamin Muller**[2]**, Jesus Villalba**[1]**, Najim Dehak**[1]**, Luke Zettlemoyer**[2]**, Gargi Ghosh**[2]**, Mike Lewis**[2]**, Srinivasan Iyer**[2†]**, Duc Le**[2†]

[1]Center for Language and Speech Processing, Johns Hopkins University
[2]Meta Superintelligence Labs
[†]Joint last author
`ylu125@jhu.edu`, `{sviyer, duchoangle}@meta.com`

## Abstract

Auto-regressive speech–text models pre-trained on interleaved text tokens and discretized speech tokens demonstrate strong speech understanding and generation, yet remain substantially less compute-efficient than text LLMs, partly due to the much longer sequences of speech tokens relative to text. This modality imbalance disproportionately allocates pre-training and inference compute to speech, potentially hindering effective cross-modal alignment and slowing performance scaling by orders of magnitude. We introduce the Latent Speech-Text Transformer (LST), which aggregates speech tokens into latent speech patches that serve as higher-level autoregressive units. This design aligns the sequence-modeling granularity between speech and text while improving computational efficiency. The resulting patches can align with textual units to facilitate cross-modal knowledge transfer and compactly capture recurring acoustic patterns such as silence. Across story-completion benchmarks under both compute-controlled and data-controlled settings, LST consistently improves speech accuracy while also improving text performance, achieving up to +6.5% absolute gain on speech HellaSwag in compute-controlled training (+5.3% in data-controlled training). Under compute-controlled scaling from 420M to 1.8B parameters in a near compute-optimal regime, gains grow with scale, and improvements persist up to 7B parameters under fixed-token budgets. These benefits extend to downstream tasks: LST stabilizes ASR adaptation and reduces the effective autoregressive sequence length during ASR and TTS inference, lowering computational cost without degrading reconstruction quality. The code is available at `https://github.com/facebookresearch/lst`.

## 1 Introduction

Inspired by the strong zero- and few-shot understanding and generation capabilities of large autoregressive textual language models with billions of parameters that are pre-trained on trillions of tokens, Lakhotia et al. (2021) introduce the task of Generative Spoken Language Modeling (GSLM) a.k.a Textless NLP, where raw speech is encoded as a sequence of discrete tokens based on a dictionary of quantized speech features, and an auto-regressive language model (LM) is trained on these tokens with Next Token Prediction (NTP). While initially successful, Cuervo & Marxer (2024) estimate that this approach would require up to three orders of magnitude more data to obtain equivalent capabilities as textual LLMs, largely owing to the same information requiring a significantly larger number of speech tokens to represent compared to text. This increased sequence length also means that these models utilize considerably more compute during inference to process the same amount of semantic content compared to text.

To improve scaling properties of large speech models by taking advantage of the comparatively larger corpus of web text compared to speech, recent efforts have leveraged transfer learning from textual modalities in the form of warm initialization from large pre-trained text models (Hassid et al., 2023), pre-training with interleaved speech-text data (Nguyen et al., 2025), and modeling

---

◇Work done while at Meta.

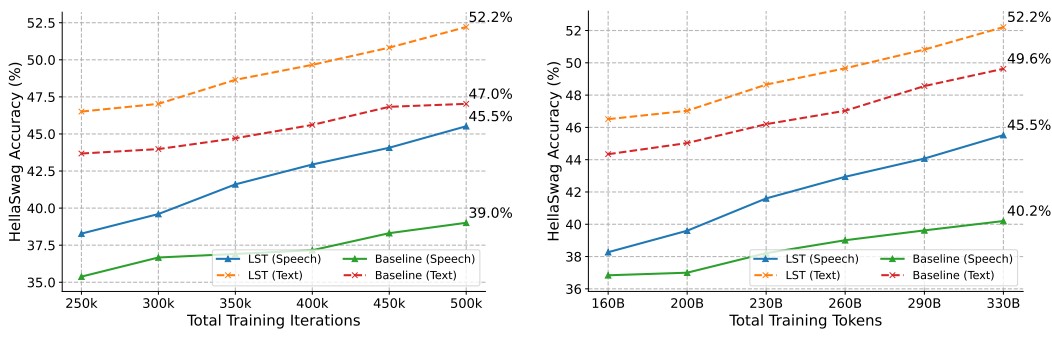

(a) *Compute-controlled*: fixed training iterations      (b) *Data-controlled*: fixed data budget

Figure 1: Comparison of LST and Baseline on HellaSwag story completion under two experimental setups, (a) *compute-controlled*: same number of training iterations and (b) *data-controlled*: same amount of training data.

speech and text in multiple streams to leverage the textual chain of thought or "inner monologue" (Défossez et al., 2024). All these works attempted to some extent to achieve *representational alignment* between text and speech, where a perfect alignment means the model can treat the two modalities interchangeably without any performance difference. Despite this, there remains a large gap between text-to-text and speech-to-speech performance on the same benchmarks, highlighting the incompleteness of the alignment. We hypothesize that the severe information-density mismatch between speech and text tokens is one of the primary factors hindering speech-text alignment.

To overcome the aforementioned challenges, we introduce the Latent Speech-Text Transformer (LST) based on the byte-latent transformer (BLT) architecture (Pagnoni et al., 2024), comprising an encoder that dynamically groups sequences of speech tokens into higher-level speech patches, a global speech-transformer that auto-regressively models interleaved sequences of textual tokens and speech patches, and a light-weight transformer decoder (Vaswani et al., 2017) that maps patches back into speech tokens of dynamic sizes. Working in terms of speech patches allows the model to encode more content given the same training cost, makes inference more efficient. These speech patches can represent higher-level speech concepts or prolonged silences, and serve to level the information density between speech and text, thus making them easier to align (see Figure 1).

We first demonstrate that LST models with fixed-size speech patching schemes similar to what Yu et al. (2023) did with text, are able to significantly outperform their non-patching counterparts. Such models are aware of the internals of patches without expending much compute in the process, in contrast with methods that expand the speech token vocabulary by applying subword tokenization, which yield poor downstream performance (Cuervo & Marxer, 2024). We further improve the performance by introducing speech-patching based on textual alignment at the word/subword levels, which crucially also includes patching large sequences of silences. Since this approach requires text-speech alignment timestamps during training and inference, we also introduce a curriculum-based method to eliminate the need for such alignments during inference.

To summarize, this paper makes the following contributions:

(1) We show that LST improves performance in both data- and compute-controlled settings compared to prior interleaved speech-text models on speech versions of popular text understanding benchmarks such as HellaSwag (Zellers et al., 2019) (see Figure 1), while substantially reducing training and inference compute and enabling efficient downstream ASR and TTS transfer.

(2) We introduce latent speech patching as a unified mechanism for compressing autoregressive speech sequences and analyze static, alignment-based, and curriculum patching strategies.

(3) We demonstrate that the benefits of LST persist and grow with scale from 1B to 7B parameters, indicating improved sample efficiency and more favorable compute-optimal scaling behavior for spoken language modeling.

Collectively, these contributions demonstrate that aligning the autoregressive modeling granularity of speech and text helps mitigate a key barrier to efficient and scalable spoken language modeling.

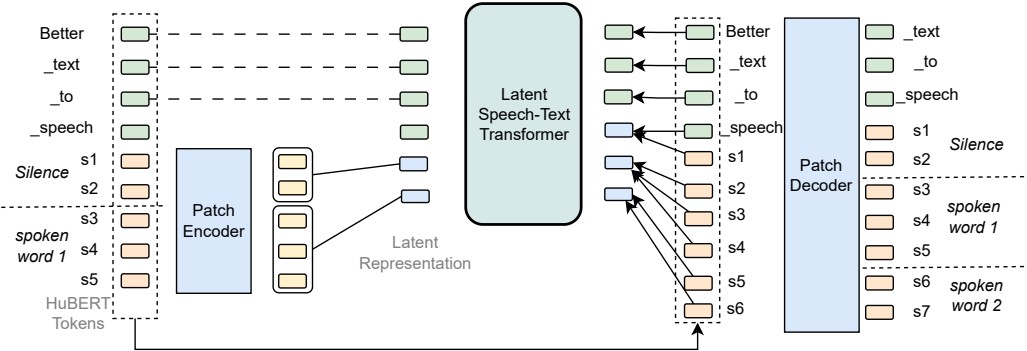

Figure 2: **Latent Speech-Text Transformer (LST).** The model encodes BPE text tokens and Hu-BERT speech tokens into a shared latent space. A *Patch Encoder* compresses local speech segments into patch representations, which are jointly processed with text tokens. A *Patch Decoder* predicts future speech tokens from latent representations, enabling alignment and transfer across modalities.

## 2 BACKGROUND

Generalized spoken language models (Lakhotia et al., 2021) typically comprise three components: (1) a speech tokenizer model that maps a raw speech waveform $s$ to a sequence of speech tokens $\{s_0, \ldots, s_n\}$, (2) a decoder-only transformer model (Vaswani et al., 2017) with parameters $\theta$ that models the distribution of the next speech token given the previous context i.e. $p_\theta(s_i|s_{<i})$, and (3) a vocoder model that maps speech token sequences back to a speech waveform, such as HiFi-GAN (Kong et al., 2020).

**Speech tokenization.** Approaches for speech tokenization include semantic tokens represented by cluster-ids obtained by k-means clustering of frame representations as in Hubert (Hsu et al., 2021), acoustic tokens obtained as discretized embeddings from residual-vector quantization bottlenecks from self-supervised neural codec models (Zeghidour et al., 2021; Défossez et al., 2024), as well as additional tokens for expressivity and also, combinations of different token categories. In this paper, we follow Hassid et al. (2023); Nguyen et al. (2025) and use Hubert tokens using a codebook of 501 speech tokens at 25Hz. Unlike Nguyen et al. (2025) we do not need to deduplicate Hubert tokens as this is organically handled by the LST architecture.

**Sequence Modeling.** Similar to LLMs for text, speech token modeling is typically done using a large transformer decoder model using causal self-attention, to maximize the likelihood of sequences from a large speech pre-training corpus ($\mathcal{D}$) in an auto-regressive fashion:

$$\mathcal{L}(\mathcal{D}; \theta) = \sum_{s \in \mathcal{D}} \sum_i \log p_\theta(s_i|s_{<i}) \tag{1}$$

**Interleaved Data.** Since speech sequences are longer and less compact that their corresponding text sequences, such models can require several orders of magnitude more data in order to achieve performance comparable to text models (Cuervo & Marxer, 2024). In order to bridge the gap, Nguyen et al. (2025) find that training on interleaved sequences of text and speech data directly correlates with improved performance. For a subset of the pre-training dataset that contains the textual sequence $\{t_0, \ldots, t_m\}$, where text tokens are obtained using a tokenizer (we use the Llama 2 tokenizer (Touvron et al., 2023) in this paper) and each text token can correspond to a span of speech tokens, the model is trained on an interleaved sequence obtained by replacing arbitrary spans of speech tokens in the sequence sequence with text tokens separated by special modality tokens. This allows the same model to be used for S→S, S→T, T→S and T→T tasks. We discuss the process of producing interleaved data from parallel text-speech data in Appendix A.1.1.

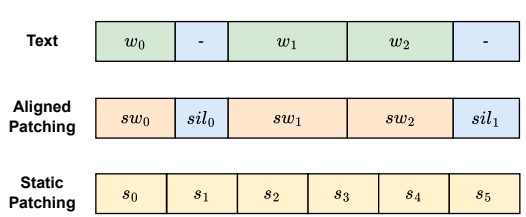
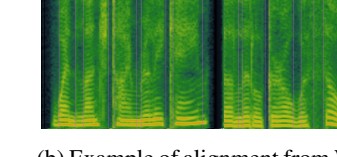

(a) Static patching segments speech into fixed-size patches, while aligned patching uses Wav2Vec2+CTC boundaries. (sw = spoken word and sil = silence.)

(b) Example of alignment from Wav2Vec2 + CTC, where purple markers indicate word boundaries, aligning the audio signal with corresponding text.

Figure 3: Illustrations of alignment and patching methods.

## 3 LATENT SPEECH-TEXT TRANSFORMERS

The core idea of the LST architecture is to auto-regressively model latent patches of tokens (using a global transformer), rather than individual tokens, similar in spirit to BLT (Pagnoni et al., 2024) which models dynamic-sized patches of bytes. The transformation of speech/text spans to patches and vice-versa takes place with the help of a light-weight patch encoder and patch decoder, and the entire model is trained end-to-end using the same token-level likelihood as before. Figure 2 illustrates this architecture specialized to the task of speech-text modeling. The majority of the compute expended in terms of FLOPs is in the global transformer, which yields savings by operating on information-dense speech patches instead of granular speech tokens. Latent patching performs bounded-context temporal aggregation, preserving phonetic discriminability while reducing sequence-level redundancy.

**Patch Encoder.** Similar to BLT, the patch encoder uses a series of sliding window self-attention and cross-attention layers to aggregate token representations into patch representations. In LST, we only patch spans of speech tokens using strategies described in Section 3.1. Note that a simple alternative to patching is to use subword tokenization methods like Byte Pair Encoding (BPE) on the speech tokens. This was also explored by Cuervo & Marxer (2024) and similar to them, failed to improve performance in our experiments (ablations in Section 5). Unlike BLT, we do not use hash embeddings, as they did not provide improvements in our experiments.

**Patch Decoder.** A light-weight transformer is used as a decoder and trained with NTP loss, with cross-attention layers inserted between every transformer layer. Each token attends to both the previously generated speech patches and text tokens to incorporate patch-level information (using cross-attention) as well as a sliding window of the past 512 tokens (using self-attention). Further architectural details of the local patch encoder and decoder, including attention roles and initialization, are provided in Appendix A.2.1.

### 3.1 PATCHING

Let $\mathbf{X} = [x_0, \ldots, x_T] \in \mathbb{R}^{T \times d}$ be speech token embeddings obtained using a learned embedding matrix applied to speech tokens $\{s_0, \ldots, s_T\}$. The process of patching maps $\mathbf{X}$ to a shorter sequence of patch embeddings $\mathbf{Z} = [z_0, \ldots, z_{T'}] \in \mathbb{R}^{T' \times d}$ by aggregating local frame segments. For a frame-index set $\mathcal{P}_i \subseteq \{0, \ldots, T\}$, a patch embedding is formed via the patch encoder:

$$z_i = \text{PatchEnc}(X_{\mathcal{P}_i}),$$

integrating the frames indexed by $\mathcal{P}_i$ into a single patch embedding. Different patching strategies correspond to different segmentation $\{\mathcal{P}_i\}$.

**Static Patching.** Speech sequence is split into non-overlapping segments of a fixed length $p$ (patch size). Each patch token is obtained by the patch encoder from the embeddings in the patch:

$$\mathcal{P}_i = \{ip, \ldots, \min((i+1)p - 1, T)\}.$$

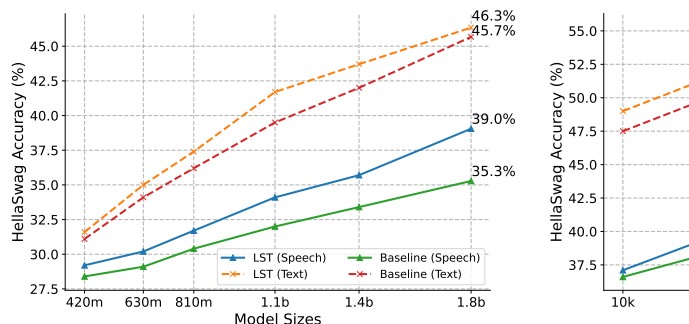

(a) *Compute-optimal scaling* (420M-1.8B). LST outperforms the baseline, with gains increasing at scale.

(b) *Sub-optimal token scaling at 7B*. Comparison at 70B tokens, below the scaling-law optimum ($\approx$140B).

Figure 4: Scaling behavior on HellaSwag (S→S and T→T.)

For $p = 3$ and input embeddings $\mathbf{X} = [x_0, x_1, x_2, x_3, x_4, x_5, x_6, \dots]$, the first patch is $\{x_0, x_1, x_2\}$, the second $\{x_3, x_4, x_5\}$, and so on. Each segment is encoded into a single patch embedding $z_i$ by the patch encoder. This provides a uniform compression ratio independent of alignment information.

**Alignment Patching.** To better synchronize speech and text at the semantic level, alignment patching leverages forced alignment timestamps between speech frames and textual units (e.g. words or BPE tokens). Let $\mathcal{A} = \{(b_k, e_k)\}_{k=1}^K$ denotes the aligned frame ranges, where $[b_k, e_k]$ spans the $k$-th textual unit. The corresponding patch is

$$\mathcal{P}_k = \{b_k, \dots, e_k\}.$$

Frames outside text spans (e.g., silence) are grouped into separate patches (Fig. 3a). If consecutive words align to $[2, 4]$ and $[6, 7]$, patches are $\{x_2, x_3, x_4\}$ and $\{x_6, x_7\}$, with silence forming $\{x_0, x_1\}$ and $\{x_5\}$. We obtain alignments with Wav2Vec2+CTC (Baevski et al., 2020), yielding one patch per text unit and silence segment (Fig. 3b). While this enforces cross-modal correspondence, it requires an auxiliary model at inference, introducing possible errors. *Curriculum patching* (Sec. 3.1) mitigates this by gradually shifting from aligned to static patching during training.

**Curriculum Patching.** Curriculum patching gradually transitions from alignment-based to static patching. Let $P(u) \in [0, 1]$ denote the probability of using alignment at training step $u$:

$$P(u) = \begin{cases} 1, & u < \tau_1, \\ 1 - \frac{u - \tau_1}{\tau_2 - \tau_1}, & \tau_1 \leq u < \tau_2, \\ 0, & u \geq \tau_2. \end{cases}$$

At step $u$, we choose alignment patches with probability $P(u)$ and static patches otherwise. This retains alignment benefits during early training while enabling simple static-only inference.

## 4 EXPERIMENTAL SETUP

### 4.1 TRAINING AND EVALUATION DATASETS

Our pre-training data comprises a mixture of text and interleaved speech datasets.

**Text.** Our text training data consists of web and academic corpora, sourced from a subset of the Llama 2 pre-training collection (Touvron et al., 2023), totaling 1.8T tokens. We follow the LLaMA 2 setup and apply its SentencePiece (Kudo & Richardson, 2018) BPE tokenizer with a 32K vocabulary.

**Speech.** Our speech training data includes speech which is discretized into HuBERT tokens (501-entry codebook at 25Hz) together with paired text transcriptions. We use LibriLight (60k hours), People's Speech (30k hours), Multilingual LibriSpeech (50k hours), and Spotify (60k hours), detailed in Table 1. All corpora are aligned using the Wav2Vec2 + CTC framework to provide token-level correspondence between speech and text (Figure 3b).

Table 1: Speech training datasets with total speech hours and the amount of Hubert tokens.

| Dataset | Hours | Hubert Tokens (B) |
|---|---|---|
| LibriLight (Kahn et al., 2020) | 44,174 | 3.7 |
| People Speech (Galvez et al., 2021) | 14,699 | 1.2 |
| Multilingual LibriSpeech (Pratap et al., 2020) | 50,601 | 4.2 |
| Spotify (Clifton et al., 2020a) | 55,309 | 4.6 |

We evaluate the model on three benchmarks, where each dataset provides a narrative context and candidate endings, and the model selects the most plausible continuation. Together, they test narrative understanding, commonsense reasoning, and topic coherence. We evaluate the model in both speech-to-speech (S→S) and text-to-text (T→T) modes. For the speech mode, we apply Kokoro TTS model (hexgrad, 2025) to generate the speech for evaluation.

**sHellaSWAG (HS).** We create a speech version of HellaSwag (Zellers et al., 2019) with Kokoro TTS. This benchmark evaluates everyday commonsense reasoning with spoken inputs and outputs. To ensure fairness, we generate the speech for prompts and responses independently and concatenate them afterwards, so that all responses are evaluated against the same speech prompt.

**StoryCloze and Topic StoryCloze (SC/TSC).** SC (Mostafazadeh et al., 2016) and its topic-based extension TSC (Hassid et al., 2023) are widely used in prior multimodal work (e.g., Nguyen et al. 2025) to test coherence and topic-sensitive reasoning. We resynthesize both datasets with Kokoro TTS for higher-quality speech inputs.

Table 2: Evaluation datasets for story completion (MC = Multiple Choice).

| Dataset | Format | Focus |
|---|---|---|
| HellaSwag (Zellers et al., 2019) | 1-in-4 MC | Commonsense reasoning |
| StoryCloze (Mostafazadeh et al., 2016) | 1-in-2 MC | Narrative coherence |
| TopicStoryCloze (Hassid et al., 2023) | 1-in-2 MC | Topic consistency |

## 4.2 LST Models and Baselines

**LST Models.** We explore four patching strategies for speech tokens:

- **Static Patching.** Fixed-length patches (4 HuBERT tokens) as in Yu et al. (2023), independent of alignment and consistent across training/inference.
- **Aligned Patching.** Uses Wav2Vec2+CTC boundaries (Fig. 3b). For each text span $[b_k, e_k]$, we form patch set $\mathcal{P}_k = \{b_k, \ldots, e_k\}$, synchronizing speech and text tokens (Fig. 3a).
- **Mixed Patching.** Randomly applies static or aligned patching per sequence, combining the robustness of static patching with the fine-grained sync of aligned.
- **Curriculum Patching.** Training shifts from aligned (first third) to mixed (middle) to static (final), leveraging early alignment while ensuring robustness to static-only inference.

**Baselines.** We include two speechLLM systems as baselines:

- **Base SpeechLLM.** Processes speech tokens directly with text tokens, without patching, similar to SpiritLM (Nguyen et al., 2025).
- **BPE SpeechLLM.** Maps speech tokens into 1k BPE units using a SentencePiece tokenizer (Kudo & Richardson, 2018) trained on 100k random speech sequences, replacing speech tokens with BPE-derived units[1].

## 4.3 Training Settings

To balance modalities, we set speech tokens to account for about one third (33%) of the total training data, while the rest (67%) is text-only. This ensures that the model benefits from large-scale text pre-

---

[1]We use the 1k configuration as our BPE baseline, as larger vocabularies (5k, 10k) showed no benefit.

Table 3: Main comparison of LST models and baselines under the **same computation budget** scheme. Each dataset reports both S→S and T→T.

| Model | Tokens (B) | | HellaSwag | | StoryCloze | | TopicStoryCloze | |
|---|---|---|---|---|---|---|---|---|
| | Int. | Text | S→S | T→T | S→S | T→T | S→S | T→T |
| Base SpeechLLM | 87 | 175 | 39.0 | 47.0 | 59.1 | 67.8 | 87.5 | 95.7 |
| BPE SpeechLLM | 95 | 190 | 38.0 | 47.5 | 58.0 | 66.4 | 87.0 | 93.5 |
| LST (Static) | 108 | 217 | 44.3 | 51.1 | 60.5 | 70.3 | 87.7 | **96.2** |
| LST (Aligned) | 108 | 217 | 42.7 | 51.7 | 60.4 | 70.4 | 86.6 | 95.7 |
| LST (Mixed) | 108 | 217 | 44.3 | 51.9 | **61.4** | 70.8 | **88.0** | 95.9 |
| LST (Curriculum) | 108 | 217 | **45.5** | **52.2** | 61.2 | **71.6** | 87.9 | 96.1 |

Table 4: Main comparison of LST models and baselines under the **same speech/text tokens** scheme. Each dataset reports both S→S and T→T.

| Model | Compute Savings | HellaSwag | | StoryCloze | | TopicStoryCloze | |
|---|---|---|---|---|---|---|---|
| | (%) | S→S | T→T | S→S | T→T | S→S | T→T |
| Base SpeechLLM | - | 40.2 | 49.6 | 60.2 | 69.1 | 87.5 | 95.2 |
| BPE SpeechLLM | 8.2% | 39.4 | 48.4 | 58.3 | 66.3 | 86.5 | 93.9 |
| LST (Static) | 19.3% | 44.3 | 51.1 | 60.5 | 70.3 | 87.7 | **96.2** |
| LST (Curriculum) | 19.7% | **45.5** | **52.2** | **61.2** | **71.6** | **87.9** | 96.1 |

training while still maintaining substantial exposure to speech for effective multimodal alignment. For comparison, SpiritLM (Nguyen et al., 2025) adopts a different composition: 33% pure speech, 33% interleaved, and 33% text tokens. Since SpiritLM starts from a text-pretrained model, the relatively smaller text fraction is sufficient. In contrast, when training from scratch, we find that using 33% interleaved and 66% text tokens yields better performance (see Appendix A.6).

## 5 RESULTS

### 5.1 PERFORMANCE UNDER CONTROLLED BUDGETS

**Compute-controlled.** We fix the number of training iterations and per-step sequence budget so that all methods process the same number of units (baseline tokens = LST patches). Table 3 shows three trends on HellaSwag. First, patching increases the effective token budget, benefiting both modalities: Curriculum Patching improves T→T by +5.2 (47.0→52.2) and S→S by +6.5 (39.0→45.5). Second, Aligned Patching is less effective at evaluation, since variable word spans often yield longer patches, reducing the test-time compute. Finally, Mixed and Curriculum patching combine the advantages of shorter evaluation patches with alignment information, consistently outperforming Static and Aligned across datasets.

**Data-controlled.** Here we fix the data budget with the same amounts of speech and text tokens. Since LST compresses sequences into patches, it processes fewer patch tokens than the baselines, leading to higher efficiency. Table 4 shows that the BPE baseline fails to surpass vanilla SpeechLLM, whereas LST continues to achieve consistent gains. On HellaSwag, Curriculum Patching improves T→T accuracy from 49.6 to 52.2 despite reduced computation, while boosting S→S from 40.2 to 45.5. Similar improvements are observed on StoryCloze and TopicStoryCloze. Overall, LST with Curriculum Patching reduces the speech–text performance gap from 9.4 to 6.7, demonstrating that alignment through patching benefits both modalities while offering meaningful compute savings.

### 5.2 SCALING BEHAVIOR

**Compute-Optimal Scaling.** Figure 4a evaluates HellaSwag accuracy under compute-optimal training from 420M to 1.8B parameters. Following Hoffmann et al. (2022), text is trained with 20× model-size tokens, while speech uses half as many tokens to preserve a 2:1 text–speech ratio.

Table 5: LibriSpeech ASR (WER) and TTS (CER) for the 1B model, reporting context units for ASR and generation units for TTS.

| Task | Model | Iters | clean↓ | other↓ | Ctx. Units | Gen. Units |
|---|---|---|---|---|---|---|
| ASR (WER %) | Baseline | 1k | 140 | 202 | 1.0× | – |
| | | 2k | 44.7 | 73.2 | | |
| | | 4k | 20.7 | 42.4 | | |
| | LST | 1k | 6.8 | **10.4** | **0.25×** | – |
| | | 2k | **6.0** | 13.3 | | |
| TTS (CER %) | Baseline | 20k | 14.1 | 15.1 | – | 1.0× |
| | LST | 20k | 14.1 | 16.2 | | **0.25×** |

At the smallest scale (420M), LST already outperforms the baseline, reaching 29.2% vs. 28.4% for speech and 31.6% vs. 31.1% for text. These improvements compound with scale: at 1.8B, LST (Speech) attains 39.0% compared to 35.3%, while LST (Text) achieves 46.3% over 45.7%. Overall, LST provides consistent gains in both modalities, with advantages apparent from the earliest scale and amplified as model capacity increases.

**Sub-Optimal Token Scaling at 7B.** Figure 4b shows training dynamics for a 7B model under a fixed processed-token budget (70B tokens), which remains below the scaling-law optimal regime ($\approx$140B). Under this sub-optimal compute setting, LST exhibits consistently faster improvement and maintains higher accuracy throughout training. Additional comparisons at 1B and 7B are summarized in Appendix A.4, where LST achieves larger gains at 1B and persistent improvements at 7B. Notably, the speech data used in our training has already been reused for multiple epochs ($\approx$6×), indicating that further token scaling becomes data-limited rather than compute-limited in this regime. The continued upward trend at 7B nevertheless suggests that training toward the scaling-law optimum with additional data would likely further amplify LST's advantage.

## 5.3 DOWNSTREAM TRANSFER AND ANALYSIS

**ASR Adaptation and Efficient TTS Generation.** To assess ASR, we fine-tune both models on LibriSpeech clean for 1k–4k iterations (batch size 4, sequence length 4096; Table 5). The baseline at 1k steps (140% / 202% WER) frequently hallucinates transcripts and produces unreliable stopping behavior, consistent with observations in (Nguyen et al., 2025). Although performance improves with iterations, it remains far worse even at 4k (>20% / 40% WER). In contrast, LST achieves 6.8% / 10.4% WER at 1k while reducing the context units during ASR inference. Under the same setup, we evaluate TTS reconstruction after 20k fine-tuning steps (Table 5). LST matches the baseline in CER and reduces the generation length by $\sim 4\times$ during TTS inference. CER is computed from Whisper-based transcriptions (Radford et al., 2023). Together, LST enables faster ASR adaptation and preserves TTS quality while requiring fewer context units and decoding steps at inference.

**Visualization of Word-Level Speech Patch Embeddings** We visualize word-level speech patch embeddings using t-SNE (van der Maaten & Hinton, 2008) (Fig. 5) from the aligned-patching LST model. Across different categories, embeddings of the same word consistently form tight clusters, while different words remain well separated. Each word forms its own cluster (e.g., he, she, they in pronouns; knife, scissors, sharpener in tools; boat, canoe, surfing in water-related terms). Related variants such as sail–sailing show stability under inflection, while semantically similar pairs like scissors–shears also appear nearby despite being distinct words. These qualitative patterns match quantitative results: within-word similarity is high ($\sim$0.87), between-word similarity is much lower ($\sim$0.43), and silhouette scores (0.65–0.68) (Rousseeuw, 1987) confirm well-separated clusters.

**Ablation on Patching Strategies.** Table 6 compares static and aligned patching. Aligned patching uses word boundaries from alignment, producing semantically coherent patches. We consider two variants: Align (sil sep.), keeping silence spans as separate patches, and Align (sil merged), merging them with adjacent words. Both outperform static patching at similar patch sizes—for instance, Align (sil sep.) reaches 60.3 on StoryCloze S→S vs. 58.7 for static size 6, and Align (sil merged) scores 38.5 on HellaSwag S→S vs. 37.2 for static size 9. *Curriculum* starts with *Align (sil sep.)* and

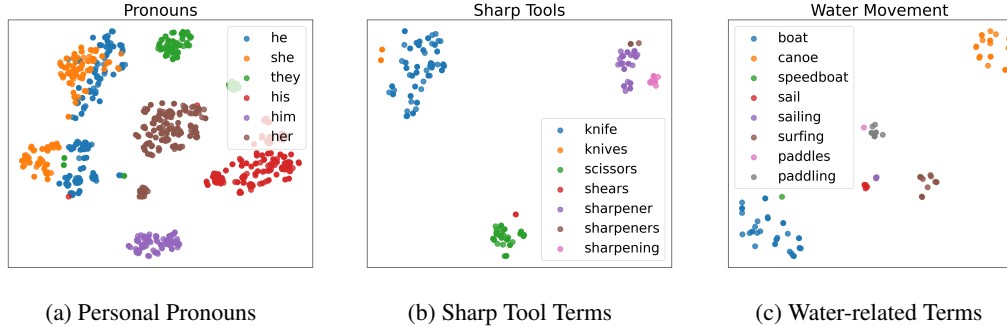

(a) Personal Pronouns           (b) Sharp Tool Terms           (c) Water-related Terms

Figure 5: Visualization of word-level speech patch embeddings from alignment patching models on HellaSwag speech, grouped by different linguistic categories. Clusters of the same word are tight and well-separated from others.

gradually shifts to *Static* during training, retaining alignment benefits while matching the shorter-patch evaluation regime; it yields the strongest and most consistent results (e.g., 41.3 on HellaSwag S→S). Overall, aligned patching better preserves semantics than static, and curriculum combines alignment supervision with static-style evaluation for the best performance. For completeness, we also report BPE-aligned patching experiments in Appendix A.7.

## 6    RELATED WORK

**LLMs using speech tokens.** Early neural audio generation methods included direct auto-regressive generation of the speech waveform (van den Oord et al., 2016), or using adversarial approaches (Kong et al., 2020). Following this, *textless NLP* work (Lakhotia et al., 2021) showed that by using discrete speech tokens obtained from self-supervised speech encoders (CPC, wav2vec 2.0, HuBERT) as targets for language modeling, can enable fully spoken LLMs. AudioLM (Borsos et al., 2023) further uses hierarchical generation, first predicts semantic tokens, and subsequent stages predict fine-grained acoustic tokens from SoundStream (Zeghidour et al., 2021), to achieve both high audio quality as well as long-term consistency. In addition to augmenting semantic speech tokens with pitch and style tokens to explicitly model expressivity, SpiritLM (Nguyen et al., 2025) also introduced interleaving speech modeling with text-tokens. More recently, Moshi (Défossez et al., 2024) propose a hierarchical *inner monologue* method, that jointly predicts time-aligned text and acoustic tokens (with distilled semantic information), together with modeling multiple-stream audio for handling full-duplex audio dialogues. Finally, similar to scaling laws for text LLMs (Hoffmann et al., 2022), Cuervo & Marxer (2024) fit scaling law curves to predict the performance of spoken LLMs, and find that they scale upto three order of magnitude more slowly than text LLMs.

**Transferring textual knowledge into speech LMs.** Slower scaling trends, together with a disproportionately lower amount of data, lead to a knowledge and reasoning gap between speech and text LLMs. To bridge this, AudioPaLM and TWIST (Rubenstein et al., 2023; Hassid et al., 2023) initialize a spoken LLM from a strong text model (PaLM-2, LLaMA), improving both speech understanding/generation and cross-lingual transfer. SpiritLM demonstrates that interleaved speech–text training significantly improves inter-modality knowledge transfer. Spectron (Nachmani et al.) uses a "Chain-of-Modality" pipeline to first produce text and then speech conditioned on the text, trading latency for stronger textual control, while Moshi (Défossez et al., 2024) uses a similar approach but generates interleaved text and speech as an inner monologue. To improve latency, LLaMA-Omni (Fang et al., 2024) style systems decode text and speech simultaneously, by upsampling textual LLM hidden states to decode speech units, before proceeding to decode the next text token.

**Speech model efficiency.** Compared to text, speech yields much longer token sequences, owing to higher frequency audio codecs, that consume many times additional compute to pre-train and generate. Efforts to mitigate this include methods to produce coarser speech units (Baade et al.; Tseng et al., 2025), hierarchical generation (Borsos et al., 2023), and producing residual tokens using parallel streams (Copet et al., 2023). Attempts at text-inspired approaches to compress token sequences such as BPE (Ren et al., 2022; Li et al., 2024) achieved limited success. In this paper, we take inspiration from recent dynamic patching approaches that have yielded improvements in other

Table 6: Comparison of patching strategies with approximately matched patch sizes. Static uses fixed patch lengths, Align (sil sep.) treats silence as separate patches, and Align (sil merged) merges silence into words, and Curriculum gradually shifts from Align to Static during training.

| Patching Strategy | Ave Patch Size (tokens) | HellaSwag | | StoryCloze | | TopicStoryCloze | |
|---|---|---|---|---|---|---|---|
| | | S→S | T→T | S→S | T→T | S→S | T→T |
| Static | 4 | 40.5 | 48.8 | 58.2 | **69.4** | 86.2 | 95.1 |
| **Curriculum (sil sep.)** | 5.8[*]→ 4 | **41.3** | **49.2** | **58.6** | 67.8 | **86.6** | **95.4** |
| Curriculum (sil merged) | 9.4 → 4 | 40.3 | 48.9 | **58.7** | 68.9 | **86.8** | **95.4** |
| **Align (sil sep.)** | 5.8[*] | **39.9** | **49.3** | **60.3** | **69.9** | 85.7 | 95.3 |
| Static | 6 | 39.4 | 49.2 | 58.7 | 69.6 | 84.9 | 94.9 |
| Static | 9 | 37.2 | **49.4** | 57.5 | **69.7** | 84.7 | 95.9 |
| **Align, (sil merged)** | 9.4 | **38.5** | 49.0 | **58.8** | **69.7** | **86.9** | **96.0** |

[*] The average patch length is 5.8 for words in Align (sil sep.), while silence has an average of 3.7.

modalities such as text (Pagnoni et al., 2024; Yu et al., 2023; Videau et al., 2025) and vision (Pang et al., 2024; Beyer et al., 2023), and extend these methods to speech-text LLMs.

**Speech Understanding Benchmarks.** Going beyond measuring only acoustic and phonetic capabilities of speech models using scores such as ABX (Kahn et al., 2020), Nguyen et al. (2020) established the Zero Resource Speech Benchmark 2021, comprising datasets/metrics to evaluate lexical (sWUGGY), syntactic (sBLIMP) and lexical-semantic (sSIMI) capabilities of spoken LLMs. Since these benchmarks contrast between very short speech segments, we found that dynamic compute approaches such as ours, do not yield significant improvements (see Appendix A.9). However, subsequently, Hassid et al. (2023) introduced the sStoryCloze and TopicStoryCloze datasets, which are story completion benchmarks in the speech modality measuring commonsense/understanding abilities of Spoken LLMs. We use these benchmarks in this paper, together with a speech version of the popular HellaSWAG textual benchmark, also measuring commonsense reasoning capabilities.

# 7 LIMITATIONS

Our study has several limitations. First, we focus on half-duplex speech–text modeling, where speech and text alternate in turns, and do not yet address full-duplex interaction required for real-time dialogue such as Moshi (Défossez et al., 2024). Second, our analysis is restricted to the pre-training stage, without exploring instruction fine-tuning or downstream adaptation, which we leave for future work. Third, some of our patching strategies, such as alignment and curriculum, rely on forced alignments during pre-training; although curriculum patching reduces this dependency at inference, fully alignment-free approaches remain an open challenge. Finally, our experiments are limited to the speech–text modality, and we have not yet extended LST to additional modalities such as image or video, which represent a promising next direction.

# 8 CONCLUSION

We presented the Latent Speech-Text Transformer (LST), a patch-based framework that aggregates speech tokens into latent units to improve computational efficiency and balance across modalities in multimodal language modeling. Across controlled-budget evaluations, scaling analyses, and downstream transfer experiments, LST consistently outperforms SpeechLLM baselines while reducing autoregressive sequence length and inference cost. Importantly, the gains arise from shortening the effective autoregressive sequence through latent speech patching, enabling more compute-efficient model scaling without sacrificing speech coverage or reconstruction quality. These findings highlight token-density imbalance as a key bottleneck in scaling spoken language models and suggest LST as a practical step toward compute-efficient unified speech–text foundation models.

ETHICS STATEMENT

This work adheres to the ICLR Code of Ethics. We use only publicly available datasets (e.g., LibriLight, People's Speech, Multilingual LibriSpeech, Spotify) and did not collect any new human-subject data. No personally identifiable information is included. Our methods aim to improve efficiency and generalization in speech–text modeling, with potential positive impacts on accessibility and multilingual applications. As with all large language models, there remain general risks of misuse, and we encourage responsible use of our work.

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

## A  APPENDIX

### A.1  DATA CONSTRUCTION AND LICENSING

#### A.1.1  INTERLEAVED DATA CONSTRUCTION

To generate interleaved sequences from parallel speech–text data, we proceed as follows:

1. **Alignment.** We obtain alignment information by using Wav2Vec2 + CTC to determine the boundaries linking text tokens to their corresponding spans of speech tokens.

2. **Span selection.** For each training example, we randomly select a contiguous span of words. The selected span is replaced by text tokens, while the following span of approximately half that length is kept as speech tokens.

3. **Modality markers.** We insert special tokens `<t>` and `` to indicate the start of text and speech segments, respectively. This ensures the model can disambiguate between modalities.

4. **Dynamic sampling.** Interleaved sequences are generated dynamically at training time, so each epoch exposes the model to different interleaving patterns for better robustness.

This process yields diverse interleaved training examples while preserving alignment between speech and text, allowing the same model to be applied uniformly to S→S, S→T, T→S, and T→T tasks.

### A.1.2 DATASET LICENSING

We summarize the licenses associated with the speech corpora used for pre-training.

- **Libri-Light.** Derived from LibriVox audiobooks that are in the public domain.
- **The People's Speech.** Includes audio released under CC BY and CC BY-SA licenses.
- **Multilingual LibriSpeech (MLS).** Distributed under CC BY 4.0.
  **Spotify Podcast Dataset.** The dataset is described by Clifton et al. (Clifton et al., 2020b), available at `https://arxiv.org/abs/2004.04270`.

## A.2 ADDITIONAL MODEL DETAILS

### A.2.1 LOCAL PATCH ENCODER AND DECODER DETAILS

We provide additional details on the local patch encoder and decoder that form the core representation transformation in LST.

**Patch Encoder.** Following the BLT-style latent aggregation paradigm (Pagnoni et al., 2024), the patch encoder converts token-level speech representations into latent patch embeddings through alternating *sliding-window self-attention* and *cross-attention* layers. Within the encoder cross-attention, latent patch queries are initialized by pooling the hidden states of speech tokens within each local span. These pooled representations then act as *queries*, while the corresponding speech token features provide the *keys and values*. Self-attention operates over a restricted temporal window to preserve local coherence while controlling computational cost, after which cross-attention aggregates the token features into a smaller set of latent patch queries that summarize contiguous speech segments. Unlike BLT, LST: (i) applies patching only to speech-token spans determined by the strategy in Section 3.1, (ii) does not employ hash embeddings, as they showed no empirical benefit in our ablations. Additionally, LST does not rely on subword tokenization (e.g., BPE applied to speech tokens), which we find provides no performance benefit in our experiments.

**Patch Decoder.** The patch decoder is implemented as a lightweight Transformer trained with next-token prediction (NTP) loss. Each decoder layer contains: (i) *causal self-attention* over a sliding window of the previously generated 512 tokens to ensure autoregressive consistency, and (ii) *cross-attention* where current token hidden states act as queries, while previously generated speech patches together with text tokens provide the keys and values. This allows token-level prediction to condition simultaneously on higher-level latent patch structure and textual context. This dual conditioning enables efficient generation while preserving long-range semantic and acoustic information encoded in the latent patches.

**Intuition.** Conceptually, the local encoder–decoder pair performs a *token → latent patch → token* information bottleneck. The encoder compresses dense speech-token sequences into structured latent summaries, while the decoder re-expands them for autoregressive prediction. This mechanism is central to LST's ability to reduce effective sequence length and computational cost without sacrificing linguistic or acoustic fidelity.

### A.2.2 MODEL ARCHITECTURE

Table 7 summarizes the hierarchical architecture used in our experiments. The model consists of a shallow patch encoder, a deep global transformer, and a moderately deep patch decoder. The local

patch modules operate with restricted attention windows to capture fine-grained context, while the global transformer uses block-causal attention with rotary position embeddings (RoPE) to handle long-range dependencies efficiently. This design balances local detail preservation with scalable long-context modeling.

Table 7: Model architecture configuration. Each module is shown with its depth, hidden dimension, number of attention heads, and other relevant settings.

| Module | Layers | Dim. | Heads | Notes |
|---|---|---|---|---|
| Patch Encoder | 1 | 1024 | 16 | Patch window = 512 |
| Global Transformer | 25 | 2048 | 16 | Block-causal; RoPE ($\theta = 5 \times 10^5$) |
| Patch Decoder | 9 | 1024 | 16 | Patch window = 512 |

### A.3 OPTIMIZATION AND TRAINING CONFIGURATION

We trained the model using the AdamW optimizer ($\beta_1 = 0.9$, $\beta_2 = 0.95$, weight decay = 0.1). The learning rate was initialized at $4 \times 10^{-4}$ and scheduled with cosine decay, including a warmup period of 2,000 steps and a minimum ratio of 0.01 at the final step. For the 1B model, training was performed on 32 H100 GPUs with a per-GPU batch size of 4 sequences (sequence length = 4,096 units), leading to a total batch size of 0.5M units. Mixed-precision training with bfloat16 was used for efficiency. Gradient clipping was applied at 1.0, and gradient accumulation was set to 1. Model parallelism used a single partition, and Fully Sharded Data Parallel (FSDP) was enabled for memory efficiency. No dropout was applied. The 1B model was trained for 200k steps, corresponding to approximately 1 trillion units, and required around 17 hours to complete on 32 H100 GPUs.

### A.4 FIXED-TOKEN SCALING ACROSS MODEL SIZES

Table 8 compares the baseline SpeechLLM and LST at 1B and 7B under an identical processed-token budget, ensuring fair cross-scale evaluation. At 1B, LST shows clear gains over the baseline (41.3 vs. 36.8 on S→S and 49.2 vs. 47.1 on T→T), indicating improved sample efficiency at smaller scale. At 7B, the advantage remains consistent, with LST achieving 44.2/55.3 compared to the baseline's 42.0/54.8. These results demonstrate that the benefits of latent speech–text modeling persist across model scales and are not limited to low-capacity regimes.

Table 8: Scaling comparison between baseline SpeechLLM and LST at 1B and 7B.

| Model | Batch (M) | Iters (k) | HellaSwag | |
|---|---|---|---|---|
| | | | S→S | T→T |
| Baseline (1B) | 0.5 | 200 | 36.8 | 47.1 |
| LST (1B) | 0.5 | 200 | **41.3** | **49.2** |
| Baseline (7B) | 4.0 | 25 | 42.0 | 54.8 |
| LST (7B) | 4.0 | 25 | **44.2** | **55.3** |

### A.5 STABILITY ANALYSIS ACROSS TASKS

We further examine the robustness of patching strategies by repeating each experiment three times and reporting the average (Ave) and standard deviation (Std). Table 9 summarizes results for HellaSwag (HS), StoryCloze (SC), and TopicStoryCloze (TSC) under both speech-to-speech (S→S) and text-to-text (T→T) directions. HellaSwag results are generally more stable than the other tasks: both Curriculum and the Baseline show near-zero std (0.13 and 0.22 for S→S), while Static is relatively less stable with larger fluctuations (0.67). By contrast, StoryCloze and TopicStoryCloze exhibit considerably higher deviations, occasionally exceeding 1.5, which indicates greater instability. Overall, Curriculum improves the average accuracy across all tasks while delivering highly consistent results on HellaSwag, underscoring its effectiveness in stabilizing training.

Table 9: Average (Ave) and standard deviation (Std) across three runs. Each task is reported with both S→S and T→T directions.

| Model | Evaluation | HellaSwag | | StoryCloze | | TopicStoryCloze | |
|---|---|---|---|---|---|---|---|
| | | Ave | Std | Ave | Std | Ave | Std |
| Curriculum | S→S | **41.4** | 0.13 | **59.2** | 0.68 | **87.1** | 0.45 |
| | T→T | **49.1** | 0.06 | **69.5** | 1.56 | **95.6** | 0.45 |
| Static (4) | S→S | 40.9 | 0.67 | 58.5 | 0.50 | 86.6 | 0.52 |
| | T→T | 48.5 | 0.37 | 69.4 | 0.11 | 95.1 | 0.19 |
| Baseline | S→S | 36.5 | 0.22 | 58.3 | 0.21 | 86.3 | 0.52 |
| | T→T | 46.3 | 0.78 | 66.6 | 1.56 | 93.9 | 1.44 |

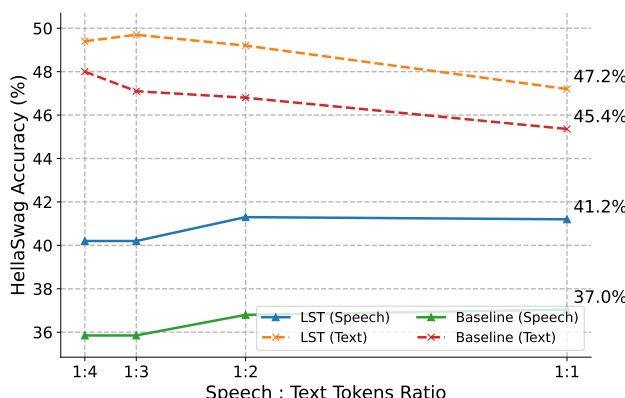

Figure 6: Effect of speech-to-text token ratio at 200k iterations. Results are reported on HellaSwag under both S→S and T→T.

## A.6    EFFECT OF SPEECH PROPORTION

Figure 6 illustrates the effect of varying the training speech–to–text token ratio on HellaSwag. Across all settings, LST consistently outperforms the baseline, and both methods exhibit the best speech–text trade-off at the 1:2 ratio. Moving from 1:3 to 1:2 improves LST (S→S) from 40.2 to 41.3 while keeping LST (T→T) high at 49.7; pushing further to 1:1 does not provide speech gain (41.2) but a large text drop (47.2, −2.5). The baseline shows the same pattern: at 1:2 it reaches 36.8 (S→S) and 47.1 (T→T), whereas 1:1 gives only 37.0 on speech (+0.2) but lowers text to 45.4 (−1.7). Averaging speech and text accuracies, the macro score peaks at 1:2 for both LST and the baseline. These results indicate that allocating about one-third of tokens to speech (1:2) offers a fair and robust operating point for both models to avoid the substantial text-side degradation seen at 1:1 while securing clear gains over lower speech ratios.

## A.7    BPE-ALIGNED PATCHING

In addition to word-aligned patching, we also explored BPE-aligned patching, where speech patches are constructed according to BPE segmentation of the text. To ensure comparability, we applied the same forced-alignment procedure at the character level and then mapped aligned spans to their corresponding BPE units. While this provides finer granularity, the resulting boundaries are less precise and the subword pieces do not always correspond to meaningful acoustic events. As shown in Table 10, word alignment generally outperforms BPE alignment in S→S (e.g., 59.4 vs. 55.6 on StoryCloze and 84.8 vs. 79.6 on TopicStoryCloze), reflecting the more reliable word-level boundaries. On the other hand, BPE achieves slightly better T→T results, likely because its patching is directly aligned with the underlying text BPE tokens. Finally, curriculum training further boosts HellaSwag S→S performance, improving from 40.0/39.2 to 41.5/41.3 for Word and BPE, respectively.

Table 10: Comparison of aligned patching strategies under a speech-to-text token ratio of 1:4. *Word Align* uses word-level forced alignment, *BPE Align* uses BPE segmentation, and *Curriculum* gradually shifts from alignment-based to static patching.

| Model | Ave Patch Size (tokens) | HellaSwag | | StoryCloze | | TopicStoryCloze | |
|---|---|---|---|---|---|---|---|
| | | S→S | T→T | S→S | T→T | S→S | T→T |
| LST (Word Align) | 5.8* | **40.0** | 49.9 | **59.4** | 68.6 | **84.8** | 94.6 |
| LST (BPE Align) | 5.0* | 39.2 | **50.1** | 55.6 | **69.1** | 79.6 | **95.6** |
| LST (Word Curr.) | 5.8→4 | **41.5** | 49.5 | 57.9 | **68.9** | 86.8 | 95.1 |
| LST (BPE Curr.) | 5.0→4 | 41.3 | 48.6 | **59.1** | 67.1 | 86.5 | **95.4** |

* The average patch length is 5.8 for words, 5.0 for BPEs, and 3.7 for silence spans.

## A.8 SC/TSC EVALUATION SETS

We additionally evaluate on the publicly available speech versions of StoryCloze (SC) and TopicStoryCloze (TSC) (Hassid et al., 2023). The released TTS audio in these benchmarks contains noticeable synthesis artifacts and unnatural prosody, which may affect evaluation reliability. To provide more realistic acoustic conditions, we generate improved TTS renderings using the same text prompts and release them for reproducibility. Table 11 reports results on both the original and improved versions. Across all four settings, LST consistently outperforms the baseline, indicating that the observed gains are robust to TTS quality variations and do not depend on a specific synthesis condition.

| Model | SC (orig.) | SC (ours) | TSC (orig.) | TSC (ours) |
|---|---|---|---|---|
| Baseline | 58.0 | 59.1 | 78.4 | 87.5 |
| LST | 60.8 | 61.2 | 79.5 | 87.9 |

Table 11: Evaluation on original and improved SC/TSC TTS sets.

## A.9 FINE-GRAINED LEXICAL AND SYNTACTIC INFORMATION

Although our main evaluation focuses on narrative understanding and commonsense reasoning, LST does not discard fine-grained lexical or syntactic information. The patch decoder preserves token-level detail through localized reconstruction, while aggregation operates only over short spans without collapsing phonetic distinctions. To verify this, we evaluate on sWUGGY and sBLIMP, which probe subword discrimination and syntactic sensitivity in spoken language modeling. As shown in Table 12, LST performs on par with the baseline, suggesting that latent patching preserves the linguistic cues required for low-level discrimination without introducing trade-offs against higher-level reasoning performance.

| Model | sWUGGY | sBLIMP |
|---|---|---|
| Baseline | 72.5 | 58.9 |
| LST | 72.8 | 59.0 |

Table 12: Fine-grained lexical and syntactic evaluation.

## A.10 SEQUENCE LENGTH, COMPUTE, AND INFERENCE SCALING

Under the compute-controlled setting, sequence composition is determined by token-level statistics of interleaved speech–text data. The data follow a fixed word ratio of 1:2 (speech : text), specified by the data construction. Because discrete speech units have higher temporal density than text tokens, this ratio corresponds in practice to approximately 0.77 speech tokens and 0.23 text tokens within the interleaved portion of the baseline sequence. LST applies 4→1 speech patching, reducing speech tokens from 0.77 to 0.19 while leaving text tokens unchanged. The total sequence length therefore

| | Interleaved Speech Data (1:2) | | Text Data | Total |
|---|---|---|---|---|
| Model | Speech units | Text units | Text units | Sequence length |
| Baseline | 0.77 | 0.23 | 2.00 | 3.00 |
| LST | 0.19 | 0.23 | 2.00 | 2.42 |
| FLOPs Reduction | | – | | $0.19\times$ |

Table 13: Token accounting under the compute-controlled setting. Interleaved speech–text data follow a 1:2 word ratio. Due to higher speech token density, the baseline allocates 0.77 speech tokens versus 0.23 text tokens within the interleaved portion. LST compresses speech tokens via 4→1 patching, reducing the overall sequence length from 3.00 to 2.42 (∼20% FLOPs reduction).

changes from 3.00 to 2.42 ($0.81\times$). Given the quadratic dependence of transformer FLOPs on sequence length, this difference corresponds to an overall compute change of approximately 20%.

During TTS inference, speech patching reduces the number of autoregressive acoustic generation steps by approximately $4\times$. Since acoustic step count dominates generation cost, this leads to a substantial reduction in overall generation computation. During ASR inference, the reduced number of acoustic sequence elements decreases the effective encoder context length by a similar factor, yielding corresponding efficiency gains in acoustic encoding.

### A.11 ALIGNER QUALITY ROBUSTNESS

Curriculum patching relies on alignment boundaries primarily during the early training stage before transitioning to static patching. To assess robustness to alignment noise, we perturb Wav2Vec2+CTC boundaries by $\pm20$–40 ms and re-evaluate alignment-based patching. This perturbation causes at most ∼1% degradation in the setting most sensitive to boundary precision (pure alignment patching). Because curriculum patching depends less on precise boundaries and gradually removes alignment during training, these results indicate that its performance benefits are expected to persist even when using simpler or less accurate aligners.

### A.12 LLM USAGE.

We used large language models solely for polishing some sentences in this paper.

