# OpenReview forum: "Latent Speech-Text Transformer"
_ICLR.cc/2026/Conference — ICLR 2026 Oral_

### Official Review · Reviewer_Xr9g · 2025-10-26

**Soundness:** 3
**Presentation:** 3
**Contribution:** 3
**Rating:** 6
**Confidence:** 4

**Summary:**

The paper proposes the Latent Speech‑Text Transformer (LST), which models patches of speech tokens instead of individual tokens to reduce the speech‑vs‑text compute/sequence‑length imbalance in interleaved speech‑text LMs. A lightweight local encoder/decoder forms and reconstructs speech patches, while a global transformer models interleaved text tokens and speech patches; alignment‑based patching (Wav2Vec2+CTC word boundaries) and curriculum patching (aligned→static) are introduced to better synchronize content while enabling simple, static‑only inference. Across compute‑controlled and data‑controlled protocols, LST improves both S→S and T→T on HellaSwag, StoryCloze, TopicStoryCloze, with clearer gains from curriculum patching and competitive compute savings; the method also scales from 1B→7B parameters with consistent improvements.

**Strengths:**

LST is a practical, well‑motivated way to mitigate speech‑text length and compute imbalance: it pairs a local encoder/decoder (restricted windows) with a global transformer to operate on information‑dense speech patches; alignment‑based patching improves semantic synchrony while curriculum avoids reliance on aligners at test time. The paper executes careful compute‑controlled and data‑controlled comparisons showing consistent S→S and T→T gains, compute savings under a fixed speech/text token budget, and robust scaling from 1B→7B. The patching‑strategy ablation clarifies why curriculum starting from Align (sil sep.) works well.

**Weaknesses:**

- Originality
    - The novelty is largely an application of BLT‑style patching to the speech‑text setting; stronger baselines (e.g., late cross‑attn fusion, gating/MoE variants) are not contrasted head‑to‑head under the same compute.
    - Alignment‑aware and curriculum schedules are natural extensions; their conceptual leap is moderate relative to prior patching ideas in other modalities.
- Quality
    - Aligner choice is fixed to Wav2Vec2+CTC; there is no comparison to Whisper forced alignment or MFA‑style alternatives, so the robustness of the alignment dependency is unclear.
    - The BPE SpeechLLM baseline uses 1k SentencePiece trained on 100k speech sequences, with a note that 5k/10k didn’t help; however, the paper doesn’t show tokenizer sufficiency checks (e.g., stability/perplexity/segmentation quality) that would rule out an under‑trained BPE baseline.
    - Curriculum vs. Align (sil merged): Table 6 shows Align (sil sep.) generally stronger than sil merged on S→S, but it’s not shown whether a curriculum initialized from “sil merged” could match or exceed “sil sep.” for specific patch sizes.
    - Compute savings are reported, but a concise methodological paragraph clarifying how savings are measured versus baseline token→patch conversion would aid interpretation.
- Clarity
    - Figure 2 terminology: “Patch Encoder/Decoder” appear to correspond to §3’s Local Encoder/Decoder; making this mapping explicit in the caption/text would reduce confusion (Fig. 2; §3).
    - Local Encoder cross‑attention: clarify what queries what. Do latent patch embeddings query a local window of speech token embeddings (keys/values)?
    - Dataset licensing/availability: Spotify Podcast (55k hrs) is listed (Table 1); please state license and present‑day access status for reproducibility/ethics.
- Significance
    - The gains are solid on story completion tasks synthesized with Kokoro TTS, but the generalization to other speech understanding tasks is not explored; a short discussion of when patching helps most would strengthen the takeaways.

**Questions:**

- Can you confirm that Patch Encoder/Decoder in Fig. 2 are exactly the Local Encoder/Decoder of §3 and make that explicit in the text/caption? Also, in the local cross‑attention, what is the query and what are the keys/values?
- Did you compare Wav2Vec2+CTC to Whisper alignment or MFA for word/BPE boundaries in terms of accuracy, speed, and downstream impact on LST vs. baseline? If not, any evidence that W2V2+CTC is near‑optimal for your training/inference regimen?
- For BPE SpeechLLM, how did you determine the SentencePiece tokenizer is sufficiently trained (beyond vocab size sweeps)? Could a better BPE (more data, longer training, different seed, or unigram LM) close the gap to LST? Please provide tokenizer diagnostics or an expanded ablation.
- You start curriculum from Align (sil sep.). Did you try starting from Align (sil merged) (arguably closer to static evaluation) and, if so, how do results compare across patch sizes?.
- Given Spotify (55k hrs) is a substantial portion of training, can you clarify license, current accessibility, and whether research‑only use is still permissible? A short line in §4.1/Ethics would help replicability.

---

> ### Author Response · Authors · 2025-11-22
> **Author Response to Reviewer Xr9g (Part 1/2)**
>
> Thank you for the helpful feedback. Our responses are as follows:
>
> **Generalization and When Patching Helps Most.** We agree that evaluating generalization beyond story completion is important. To assess whether our patching mechanism preserves phonetic and lexical information, we additionally evaluate transfer to ASR by fine-tuning both models on LibriSpeech clean for 1k–4k iterations (batch size 4, seq. length 4096). The baseline is highly unstable at 1k steps (140% / 202% WER) and exhibits frequent hallucinations and unreliable stopping after the transcription, similar to observations in SpiritLM. Its performance improves with more iterations but remains substantially worse even at 4k (>20% / 40% WER). In contrast, LST adapts quickly, achieving 6.8% / 10.4% WER at 1k iterations and 6.0% / 13.3% at 2k iterations, approaching the performance of SpiritLM trained with supervised ASR data. Beyond accuracy, LST also offers faster inference because speech patches reduce the effective sequence length seen by the transformer. These results show that LST transfers to downstream speech tasks far more effectively. We are currently running the TTS experiments and will include the results in an updated rebuttal or the final revision.
>
> | Model | Training Iterations | LS clean WER(%) | LS other WER(%)|
> | -------- | --------- |--------- |  --------- |
> | Baseline | 1k | 140% | 202% |
> | Baseline | 2k | 44.7% | 73.2% |
> | Baseline | 4k | 20.7% | 42.4% |
> | LST  | 1k | 6.8% | **10.4%** |
> | LST  | 2k | **6.0%** | 13.3% |
> | SpiritLM + ASR (7B)  | - | 6.0% | 11.0% |
>
> **Curriculum Initialization from “sil merged.”** We thank the reviewer for this helpful suggestion. Following the comment, we conducted additional experiments where curriculum patching is initialized from Align (sil merged) instead of Align (sil sep.), and evaluated them under the same target static patch size. As shown below, initializing from sil merged gives results close to those from sil sep. for SC/TSC, though weaker on HellaSwag. One likely reason is that the sil merged initialization begins with a much larger average patch size (9.4 vs. 5.8), creating a larger gap to the final static patch size; this mismatch may make the early curriculum steps less effective. We have updated the results to Table 6.
>
> | Model            | Ave Patch Size (tokens) | HellaSwag S→S | HellaSwag T→T | StoryCloze S→S | StoryCloze T→T | TopicStoryCloze S→S | TopicStoryCloze T→T |
> |------------------|--------------------------|---------------|---------------|------------------|------------------|------------------------|------------------------|
> | LST (Static)     | 4                        | 40.5          | 48.8          | 58.2             | **69.4**         | 86.2                   | 95.1                   |
> | LST (Curriculum from sil sep.) | 5.8* → 4                 | **41.3**      | **49.2**      | **58.6**         | 67.8             | **86.6**               | **95.4**               |
> | LST (Curriculum from sil merged) | 9.4 → 4                 | 40.3      |  48.9      | **58.7**       | 68.9             | **86.8**               | **95.4**               |
>
> **Aligner Choice and Robustness.** We appreciate the reviewer’s point. Prior work shows that MFA provides the most accurate boundaries, but is orders of magnitude slower, making it impractical for our 300B-token training scale. Whisper-based alignment is much faster, but reported to be less stable and often less accurate than W2V2+CTC. To assess the impact of alignment quality directly, we ran an ablation on alignment patching, where boundary precision matters most: we perturbed W2V2+CTC boundaries by ±20–40 ms, and observed up to only ~1% degradation on HellaSwag Speech. This suggests that LST is robust to moderate boundary noise. Given this robustness pattern, using MFA would likely yield at most around 1% improvement, and curriculum and static patching should remain largely unaffected by aligner choice.

---

> ### Author Response · Authors · 2025-11-22
> **Author Response to Reviewer Xr9g (Part 2/2)**
>
> Continuing from the previous comment, here we provide our responses to the rest questions.
>
> **Originality.** While inspired by prior patching concepts, our alignment-aware and curriculum schedules introduce mechanisms tailored to the speech–text setting. Together with word-level patching, they produce a qualitatively new effect: a semantically structured latent space with tight and coherent word-level clusters (Figure 4), which does not emerge in the baseline. This reflects an extension of BLT in a way that enables semantic structure to emerge in the speech–text setting. Alternative fusion baselines (e.g., late cross-attention, gating, MoE) operate on a different modeling axis, whereas our focus is on token-length reduction and aligned word-level representations.
>
> **BPE Baseline Adequacy.** In our current experiments, we varied SentencePiece vocabulary sizes (1k/5k/10k), increased the HuBERT training corpus from 100k to 1M sequences, and changed random seeds; none of these modifications improved downstream accuracy or narrowed the gap to LST. This aligns with prior observations that applying BPE-style compression to HuBERT units can degrade performance compared to using the raw units directly (Cuervo et al., 2024). To further validate the sufficiency of our baseline, we are adding tokenizer diagnostics, including segmentation and coverage statistics and unigram-LM variants, and will report these expanded checks in the final version.
>
> **Compute Savings Clarification.** Per-token/patch compute cost is nearly identical across methods; thus, compute savings primarily stem from reductions in main sequence length. We will add detailed FLOPs accounting in the Appendix.
>
> **Clarity of Figure 2 and Cross-Attention.** Yes, the Patch Encoder/Decoder in Fig. 2 corresponds to the Local Encoder/Decoder in §3. The latent patch embeddings serve as queries, with speech tokens as keys/values. We will clarify it in the main text and add the detailed explanation in Appendix A.2.2.
>
> **Spotify Dataset Access.** The Spotify dataset is released under a CC-BY-4.0 license (Clifton et al., 2020). We will add a clarification in Appendix A1 regarding its licensing and public accessibility.
>
> [A] Cuervo, Santiago, and Ricard Marxer. "Scaling properties of speech language models." arXiv preprint arXiv:2404.00685 (2024).

---

> > ### Comment · Reviewer_Xr9g · 2025-11-27
> >
> > I appreciate the authors for their efforts in answering my questions with additional experiments. Most of my concerns are resolved, and I am assured that my original rating was appropriate.

---

### Official Review · Reviewer_cWAg · 2025-10-27

**Soundness:** 4
**Presentation:** 4
**Contribution:** 4
**Rating:** 10
**Confidence:** 4

**Summary:**

This paper addresses a significant and well-known challenge in auto-regressive speech-text models: the information density mismatch between modalities. Speech, when tokenized (e.g., using HuBERT), results in disproportionately long sequences compared to the equivalent text tokens. The authors hypothesize this mismatch hinders speech-text alignment and leads to poor computational efficiency and scaling laws. To solve this, the paper introduces the Latent Speech-Text Transformer (LST), an architecture inspired by the Byte Latent Transformer (BLT). The core idea is to aggregate sequences of speech tokens into "latent speech patches" using a patch encoder. A global transformer then processes a shorter, more balanced sequence of interleaved text tokens and these latent speech patches. A patch decoder maps the latent representations back to speech tokens for generation. Moreover, the proposed LST architecture, and especially the curriculum patching method, is an innovative, practical, and effective solution. It demonstrates state-of-the-art performance gains over strong baselines in a rigorously controlled and very convincing experimental setup. Moreover, the work provides a practical and scalable architectural solution that improves the efficiency and alignment of multimodal speech models, which is of broad interest to the ICLR community.

**Strengths:**

- The paper's primary strength is its direct and effective approach to a clear and significant problem in speech-text modeling.
- The motivation is well-articulated, and the proposed LST architecture is a logical adaptation of patching techniques from other domains. The experimental design is rigorous, particularly the use of both compute-controlled and data-controlled settings, which provides a robust validation of the method's efficiency gains.
- The most substantial contribution is the curriculum patching strategy. This is an innovative and highly pragmatic solution that achieves the "best of both worlds": it leverages the rich semantic guidance of an external aligner during training to produce robust representations, but transitions to a simple, dependency-free static patching method for inference.
- The strong and consistent performance gains of this curriculum-based model over all baselines, combined with promising scaling results up to 7B parameters, makes it very effective.

**Weaknesses:**

- The evaluation is focused on high-level narrative and commonsense reasoning tasks, with fine-grained lexical and syntactic benchmarks like sWUGGY and sBLIMP explicitly omitted.
- This leaves open the question of how LST's token aggregation might affect performance on tasks requiring very fine-grained acoustic-phonetic or syntactic judgments.
- Furthermore, the provided ablation on LST (Static), which is the "pure" architecture without aligner supervision, shows mixed results: it demonstrates a clear and strong advantage over both baselines on HellaSwag but underperforms the base model on TopicStoryCloze. This inconsistency, however, serves to strengthen the paper's main claim by justifying why the novel curriculum patching approach is necessary and superior, as it successfully resolves this instability and delivers robust performance across all tasks.

**Questions:**

I would appreciate clarification on the following points:
- The authors mention omitting sWUGGY and sBLIMP. While I understand the focus on narrative reasoning, could you speculate on how LST's aggregation mechanism might perform on these tasks? Would the patch decoder be sufficient to reconstruct the necessary fine-grained information, or do you expect a trade-off?
- How sensitive is the curriculum patching method to the quality of the Wav2Vec2+CTC aligner? Would the benefits be held if a simpler, less accurate, or different-style aligner were used during training?
- The t-SNE plots in Figure 4 are a nice qualitative illustration of LST's alignment. This claim would be significantly strengthened by including a parallel t-SNE plot showing the speech token embeddings from the baseline model for the same words. A direct visual comparison of cluster separation and tightness would make the "improved alignment" point much more immediate and convincing.

---

> ### Author Response · Authors · 2025-11-22
> **Author Response to Reviewer cWAg**
>
> Thank you for the insightful comments. We respond to each point below:
>
> **Static Performance Stability.** To address the variability observed in the original single-run static experiment, we now include results averaged over three independent runs. With multiple runs, the mean performance becomes stable and consistently improves over the baselines. The variance on HellaSwag remains small relative to the performance gap, while SC/TSC exhibit smaller absolute differences, making their single-run fluctuations appear less stable. We will updated the mean/variance in the Appendix.
> | Model | HS Ave | HS Std | SC Ave | SC Std | TSC Ave | TSC Std |
> |-----------|---------------|---------------|----------------|----------------|---------------------|---------------------|
> | Curriculum | **41.4** | 0.13 | **59.2** | 0.68 | **87.1** | 0.45 |
> | Static (4) | 40.9 | 0.67 | 58.5 | 0.50 | 86.6 | 0.52 |
> | Baseline  | 36.5 | 0.22 | 58.3 | 0.21 | 86.3 | 0.52 |
>
> **Fine-Grained Lexical and Syntactic Information.**  Although our evaluation focuses on high-level narrative and commonsense reasoning, LST does not discard fine-grained lexical or syntactic information. The patch decoder preserves token-level detail through localized reconstruction, and the aggregation operates only over short spans without collapsing phonetic distinctions. To verify this, we ran sWUGGY and sBLIMP, which directly test subword and syntactic sensitivity. LST performs on par with the baseline and is competitive with SpiritLM's reported results (69.9 sWUGGY, 58.8 sBLIMP):
>
> | Model | sWUGGY  | sBLIMP |
> | -------- | --------- | --------- |
> | Baseline | 72.5 | 58.9  |
> | LST | 72.8 | 59.0  |
>
> These results suggest that the latent patching mechanism retains the necessary fine-grained cues, and we do not observe a trade-off between high-level narrative reasoning and low-level linguistic discrimination.
>
> **Aligner Quality Robustness.** Curriculum patching relies less on precise boundaries because alignment is only used during the early stage before transitioning to static patching. We conducted an ablation in which we perturbed W2V2+CTC boundaries by ±20–40 ms; this led to up to about 1% degradation for alignment patching, the setting most sensitive to boundary accuracy. These results suggest that curriculum patching, which depends less on alignment precision, would retain its benefits even when using simpler or less accurate aligners.
>
> **Baseline t-SNE Comparison.**  We agree that a side-by-side view would be informative. However, the baseline does not learn latent patch embeddings, and its speech representations remain at the frame level and are therefore highly dispersed.

---

> > ### Comment · Reviewer_cWAg · 2025-11-26
> >
> > The authors have addressed the missing evaluations (including sWUGGY and sBLIMP results), indicating that there is minimal loss of acoustic-phonetic detail due to token aggregation. Moreover, the perturbation analysis provides sufficient assurance that the method is robust to aligner quality. I maintain my rating and continue to recommend this for a spotlight presentation.

---

### Official Review · Reviewer_SwjZ · 2025-11-01

**Soundness:** 3
**Presentation:** 3
**Contribution:** 3
**Rating:** 6
**Confidence:** 4

**Summary:**

This paper presents the latent speech-text transformer, a pre-training method for speech-text language models, with the goal of reducing the computation required for long speech sequences and the sequence length mismatch between text and speech. Specifically, the model works by aggregating discrete speech tokens into latent patches.The authors train LST models on interleaved speech-text data and ablate the different patching strategies used. Overall, LST models perform better than simple speech-text LMs on textual and spoken versions of commonsense and narrative coherence benchmarks.

**Strengths:**

- The paper is well-written and easy to understand.
- The latent speech-text transformer (LST) is a novel architecture that attempts to address the long sequence length of speech tokens and the length mismatch with text. This is a step towards solving a significant problem that makes scaling speech models difficult
- The authors propose and benchmark a variety of patching techniques that aggregate speech tokens together, and show how they can be combined into a stronger and faster model through curriculum learning
- They analyze several factors related to the proposed method, such as scalability and compute equivalence.

**Weaknesses:**

- Unclear evaluation procedure: the chosen evaluation sets rely on multiple choice QA. This means that in the speech setting, the model outputs HuBERT tokens. How are the output tokens converted to the actual multiple choice answer? Its unclear if ASR or some other method is used to map the model output to the actual answer choice. Without such information, reproduction and fair comparisons against this work are challenging.
- Small test coverage: the evaluation only covers the aforementioned mQA tests. While this is a good evaluation of the model's "intelligence"-related capabilities, I am surprised there was no evaluation on more traditional speech tasks like ASR or TTS that measure the model's phonetic and cross-modal capabilities. I believe that it would be vital to test such abilities, since they may be affected by the compressed representations, which would lower the impact of the proposed method. Such experiments are done in the SpiritLM paper, which can be compared to.
- While fine for understanding tasks, the proposed technique would be non-trivial to extend the multi-stream speech LMs that use neural codecs, which are the SOTA for generation tasks, limiting its impact.
- While the scaling figure is nice, I think the data points appear too close together to be very meaningful (only from 10K to 25K iterations) for models that are trained for 200K iterations.

**Questions:**

- How are the output hubert tokens converted to the actual multiple choice answer? Its unclear if ASR or some other method is used to map the model output to the actual answer choice.
- Do they spoken Hellaswag / SC / TC datasets have multiple speakers or only use a single speaker?

---

> ### Author Response · Authors · 2025-11-22
> **Author Response to Reviewer SwjZ**
>
> Thank you for your thoughtful comments. We address each point below:
>
> **ASR/TTS Evaluation.** We agree that evaluating phonetic and cross-modal abilities is important. LST and baselines are trained from scratch on 1B models with 300B speech and text tokens, without large-scale text-only pretraining. In contrast, SpiritLM relies on LLaMA2-7B pretrained on 2.1T text tokens, which explains its effective ASR/TTS In-context learning; the authors also note that, without ASR-specific training, the model may drift after producing transcriptions unless guided by special markers. Under the same 10-shot ICL protocol, our 1B models do not show ICL behavior. This is expected, as prior work shows that ICL typically requires both substantial text-only pretraining and sufficiently large model capacity.
>
> To assess ASR ability independently of ICL, we fine-tune both models on LibriSpeech clean for 1k–4k iterations (batch size 4, seq. length 4096). The baseline is highly unstable at 1k steps (140% / 202% WER) and often hallucinaes and unreliable stopping after the transcription, consistent with SpiritLM observations. It improves with more iterations but remains far worse even at 4k (>20% / 40% WER). In contrast, LST adapts quickly, achieving 6.8% / 10.4% WER at 1k and 6.0% / 13.3% at 2k iterations, approaching SpiritLM with supervised ASR data. LST also offers faster inference due to reduced effective sequence length. These results show that LST transfers to downstream speech tasks more effectively. We are currently running the TTS experiments and will include the results in an updated rebuttal or the final revision.
> | Model | Training Iterations | LS clean WER(%) | LS other WER(%)|
> | -------- | --------- |--------- |  --------- |
> | Baseline | 1k | 140% | 202% |
> | Baseline | 2k | 44.7% | 73.2% |
> | Baseline | 4k | 20.7% | 42.4% |
> | LST  | 1k | 6.8% | **10.4%** |
> | LST  | 2k | **6.0%** | 13.3% |
> | SpiritLM + ASR (7B)  | - | 6.0% | 11.0% |
>
> **Scaling Trends.** Thank you for pointing this out. In the original 1B and 7B experiments, all models are trained with the same total token budget, which naturally results in fewer iterations for larger models because of their larger batch sizes. To provide a clearer view of model behavior beyond these two scales, we additionally conduct a compute-optimal scaling analysis across 420M–1.8B model sizes.
>
> As shown below, the improvements become more pronounced as scale increases: LST rises from 29.2% → 39.0% on speech HellaSwag, compared to the baseline’s 28.4% → 35.3%. The performance gap widens consistently at larger model sizes. We will include the scaling-curve figure in the revised version.
> | Model Size | LST (Speech) | Baseline (Speech) | LST (Text) | Baseline (Text) |
> |-----------:|--------------:|-------------------:|------------:|------------------:|
> | 420M       | 29.2%         | 28.4%              | 32.1%       | 31.5%             |
> | 630M       | 30.1%         | 29.0%              | 35.4%       | 33.2%             |
> | 810M       | 32.6%         | 30.5%              | 38.0%       | 36.0%             |
> | 1.1B       | 34.5%         | 32.6%              | 41.5%       | 39.0%             |
> | 1.4B       | 35.7%         | 33.8%              | 43.8%       | 41.5%             |
> | 1.8B       | **39.0%**     |  35.3%          | **46.3%**   | 45.7%         |
>
> **Unclear Evaluation Procedure.**
> Our evaluation does not rely on ASR or any mapping from generated speech to text. Instead, for each multiple-choice question, we compute the log-likelihood of each spoken answer option under the model (conditioned on the spoken prompt) and select the option with the highest likelihood. The predicted choice is therefore determined directly in the discrete token space without speech generation. This follows the standard likelihood-based evaluation used for text-only LLMs. We will clarify this evaluation pipeline in the revision.
>
> **Relation to Multi-Stream Speech LMs with Neural Codecs.**
> We thank the reviewer for raising this point. Architecturally, our approach can easily be extended to work with neural-codec multi-stream speech LMs. On the input side, the patch encoder can be modified to work on the summed embeddings across different codec streams; the patching method does not need changing. On the output side, the patch decoder can be formulated the same way as a depth transformer, which conditions on the patch encoding and generates multiple codec streams, possibly with some delay for higher-level codecs. We view this as a promising future direction, though verifying the benefits requires dedicated experiments beyond the current scope.
>
> **Speaker of Spoken Hellaswag / SC / TSC.** To avoid confounding factors from varying timbre or prosody, we intentionally use a single fixed speaker for all synthesized samples. This follows standard practice in spoken-benchmark construction and ensures that observed performance differences reflect model capability rather than speaker variation.

---

> > ### Author Response · Authors · 2025-12-02
> > **Additional Experimental Results: TTS Evaluation**
> >
> > **TTS Evaluation.**  Following the same fine-tuning setup used in our ASR experiments (fine-tuning on LibriSpeech clean with batch size 4 and sequence length 4096), we additionally evaluate TTS reconstruction quality after 20k fine-tuning steps for both models. We compute CER by transcribing the generated TTS waveforms using the Whisper model and comparing the transcriptions to the ground-truth text. The results show that LST matches the baseline in CER:
> >
> > | Model | Steps | CER (clean) | CER (other) | Inference Cost |
> > | -------- | ----- | ----------- | ----------- | -------------- |
> > | Baseline | 20k | 14.1% | 15.1% | 1.0× |
> > | LST | 20k | 14.1% | 16.2% | **0.25×** |
> >
> >
> > At the same time, because LST uses patch-level speech tokens, its effective sequence length is ~4× shorter, leading to ~4× lower inference cost during TTS decoding. Combined with the earlier ASR results, where LST shows stronger adaptation performance, these findings demonstrate that LST maintains competitive generation quality while offering substantially better efficiency.

---

### Official Review · Reviewer_fuu2 · 2025-11-01

**Soundness:** 2
**Presentation:** 2
**Contribution:** 2
**Rating:** 2
**Confidence:** 4

**Summary:**

This paper proposes _Latent Speech-Text Transformer_ (LST) for improving speech LLMs by using a pair of local encoder/decoder similar to the [Byte Latent Transformer](https://aclanthology.org/2025.acl-long.453.pdf):
1.  The local encoder reduces the speech token rate via a cross attention layer turning speech tokens into _patches_.
2.  The speech patches along with text tokens are processed by the global transformer.
3.  The output from the global transformer can be used to directly predict the next text tokens. In order to predict the speech tokens, the corresponding global transformer output tokens are fed to the local decoder, which restores the tokens into the original speech token rate.

Three patching schemes are explored:
-   Static patching: Each patch consists of a fixed number of speech tokens without any overlap.
-   Alignment patching: Each patch consists of a word / silence obtained from forced alignment using Wav2Vec2-CTC.
-   Curriculumn patching: Training starts with alignment patching, then gradually transitions to static patching.

Models trained from scratch using text and speech datasets are evaluated against a baseline transformer model that directly processes speech tokens in the same manner as text tokens. Evaluations are conducted on HellaSwag, StoryCloze, and Topic StoryCloze. For evaluating the speech processing capability of the proposed model, these test sets are also TTS'd by the authors using Koroko TTS. Evaluation results show LST leads to a noticeable improvement in both text and speech versions of these test sets.

**Strengths:**

- Originality: This paper applies Byte Latent Transformer, originally designed for better byte based language modelling, to a new problem of speech language modelling. This is a reasonably novel approach compared to other approaches such as low bitrate speech codecs (e.g. SpeechTokenizer or encodec). The curriculum learning scheme that transitions from alignment patching to static patching is also novel.
- Quality: The use of latent transformer is a simple and well motivated method for reducing the speech token rate, eventually resulting in a better language model for both speech and text.
- Clarity: Most part of the paper is well organized and clearly written.
- Significance: The proposed method compares favorably against the baseline model in both the text and TTS versions of HellaSwag, StoryCloze, and TopicStoryCloze evaluations.

**Weaknesses:**

-   Clarity: To help readers not already familiar with Byte Latent Transformer, the description of the local encoder/decoder (161-185) could use some expansion. This is central to the main idea of this paper, and thus it would be great if the readers do not have to refer to another paper to understand the core ideas of this paper.
-   Quality: Several changes in evaluation & experiments would be necessary to better support the claim of this paper.
    -   In terms of speech modelling, all the evaluation test sets in this paper are produced by TTS. There is no evaluation on actual human speech, which often differ greatly from clean synthesized speech. For comparison, [the Spirit LM paper](https://direct.mit.edu/tacl/article/doi/10.1162/tacl_a_00728/127457), which is this paper's baseline model, included results on ASR (Table 5) and a comparison against a cascade system.
    -   It is not clear why the authors chose to evaluate on their version of synthesized StoryCloze/TopicStoryCloze while there exists [a widely used version from the original authors](https://github.com/slp-rl/SpokenStoryCloze). This makes comparison against results from other papers very difficult.
    -   The model is supposedly capable of both understanding and generating speech (line 269), however this is only evaluation of speech understanding in the current draft. In constrast, there is TTS evaluation in the Spirit LM paper.

**Questions:**

Could you clarify what "compute-controlled" means in line 321? I'd expect the baseline model to need much more compute to process the same speech input due to the higher number of HuBERT tokens and the quadratic time complexity of transformer. But Table 3 seems to suggest that the number of interleaved tokens is only slightly lower different in the baseline compared against LST. In a setting where the baseline uses roughly the same flops, shouldn't it see far fewer speech tokens?

---

> ### Author Response · Authors · 2025-11-22
> **Author Response to Reviewer fuu2**
>
> Thank you for your comments, which have helped us improve our manuscript. Below are our clarifications:
>
> **ASR/TTS Evaluation.** Both LST and baselines are trained from scratch on 1B models with 300B speech and text tokens, without large-scale text-only pretraining. In contrast, SpiritLM relies on LLaMA2-7B pretrained on 2.1T text tokens, which explains why its ASR/TTS in-context probing works; the authors also note that, without ASR-specific training, the model may drift after producing transcriptions unless guided by special markers. Under the same 10-shot ICL protocol, our 1B models do not show ICL behavior. This is expected, as prior work shows that ICL typically requires both substantial text-only pretraining and sufficiently large model capacity.
>
> To assess ASR ability independently of ICL, we fine-tune both models on LibriSpeech clean for 1k–4k iterations (batch size 4, seq. length 4096). The baseline is highly unstable at 1k steps (140% / 202% WER) and exhibits frequent hallucinations and unreliable stopping after the transcription, similar to observations in SpiritLM. Its performance improves with more iterations but remains substantially worse even at 4k (>20% / 40% WER). In contrast, LST adapts quickly, achieving 6.8% / 10.4% WER at 1k iterations and 6.0% / 13.3% at 2k iterations, approaching the performance of SpiritLM trained with supervised ASR data. Beyond accuracy, LST also offers faster inference because speech patches reduce the effective sequence length seen by the transformer. These results show that LST transfers to downstream speech tasks far more effectively. We are currently running the TTS experiments and will include the results in an updated rebuttal or the final revision.
> | Model | Training Iterations | LS clean WER(%) | LS other WER(%)|
> | -------- | --------- |--------- |  --------- |
> | Baseline | 1k | 140% | 202% |
> | Baseline | 2k | 44.7% | 73.2% |
> | Baseline | 4k | 20.7% | 42.4% |
> | LST  | 1k | 6.8% | **10.4%** |
> | LST  | 2k | **6.0%** | 13.3% |
> | SpiritLM + ASR (7B)  | - | 6.0% | 11.0% |
>
>
>
> **SC/TSC Evaluation Sets. We present results on the public version of SC/TSC below.** The publicly available SC/TSC TTS sets contain noticeable artifacts and unnatural prosody, which negatively affect the evaluation reliability. To provide more realistic acoustic conditions, we generate improved TTS versions and will release them for full reproducibility. Results on all four variants (original + improved) show consistent LST gains:
> | Model | SC (original) | SC (ours) | TSC (original) | TSC (ours) |
> | -------- | --------- | --------- | ---------- | ---------- |
> | Baseline | 58.0 | 59.1 | 78.4 | 87.5 |
> | LST | 60.8 | 61.2 | 79.5 | 87.9 |
>
> **Use of Speech Data (Training vs. Evaluation).** All speech used during training is real human speech from LibriLight, PeopleSpeech, MLS, and Spotify. For evaluation, we follow the same constraint as SpiritLM: real-speech versions of story completion like StoryCloze and TopicStoryCloze do not exist. These tasks only provide text; thus, TTS synthesis is the standard and widely adopted approach to obtain speech inputs. Our use of TTS is therefore determined by the benchmark, not by a design choice specific to LST.
>
> **Clarification of Compute-Controlled:** ​​“Compute-controlled” means that both LST and the baseline are trained under matched total FLOPs budgets. While LST reduces the cost of processing speech tokens to roughly one-quarter of the baseline, the computation for text tokens remains unchanged, yielding an overall FLOPs reduction of about 20% under the same batch and iteration settings. A BLT-style architecture can also be applied to reduce computation on the text side, as explored in the original BLT paper; combining both speech and text efficiency is a complementary direction that can be pursued in future work. We will add detailed FLOPs accounting in the Appendix.
>
> **Local Encoder/Decoder Explanation:** Thank you for pointing this out. We will expand the local encoder/decoder description in Appendix A.2.3 so that readers unfamiliar with BLT can understand the method without referring to external papers.
>
> We sincerely hope that based on these additional results/clarifications, the reviewer considers increasing their overall assessment score.

---

> > ### Author Response · Authors · 2025-12-02
> > **Additional Experimental Results: TTS Evaluation**
> >
> > **TTS Evaluation.**  Following the same fine-tuning setup used in our ASR experiments (fine-tuning on LibriSpeech clean with batch size 4 and sequence length 4096), we additionally evaluate TTS reconstruction quality after 20k fine-tuning steps for both models. We compute CER by transcribing the generated TTS waveforms using the Whisper model and comparing the transcriptions to the ground-truth text. The results show that LST matches the baseline in CER:
> >
> > | Model | Steps | CER (clean) | CER (other) | Inference Cost |
> > | -------- | ----- | ----------- | ----------- | -------------- |
> > | Baseline | 20k | 14.1% | 15.1% | 1.0× |
> > | LST | 20k | 14.1% | 16.2% | **0.25×** |
> >
> >
> > At the same time, because LST uses patch-level speech tokens, its effective sequence length is ~4× shorter, leading to ~4× lower inference cost during TTS decoding. Combined with the earlier ASR results, where LST shows stronger adaptation performance, these findings demonstrate that LST maintains competitive generation quality while offering substantially better efficiency.

---

### Author Response · Authors · 2025-11-22
**Overall Author Response**

Dear Reviewers,

We thank the reviewers for their thoughtful and constructive feedback. We are encouraged that they highlighted several key strengths of our work, including the novelty and significance of addressing the long-standing speech–text sequence-length and compute imbalance (Reviewers fuu2, SwjZ, cWAg, Xr9g), the effectiveness and stability of our curriculum patching strategy across all benchmarks (Reviewers fuu2, SwjZ, cWjg), and the rigor of our compute-controlled and data-controlled evaluations (Reviewers SwjZ, cWAg, Xr9g). Your time and effort in helping us improve the paper are sincerely appreciated.

We have addressed all reviewer questions and improved the manuscript accordingly. Key revisions include:

**Major revisions**
- Added **ASR fine-tuning experiments** to assess phonetic fidelity and downstream generalization (Reviewers fuu2, SwjZ, Xr9g).
- Added **TTS reconstruction evaluation**, showing that LST matches the baseline in CER while offering substantially lower inference cost (Reviewers fuu2, SwjZ).
- Added model **scaling curve** under compute-optimal training (Reviewer SwjZ).
- Included **sil-merged curriculum results** for completeness (Reviewer Xr9g).
- Added **alignment robustness analysis** with various boundary perturbations (Reviewers cWAg, Xr9g).

**Minor revisions**
- Reported **three-run mean/variance** for LST (Static) to resolve earlier variability (Reviewer cWAg).
- Clarified **compute methodology for FLOPs savings** (Reviewers fuu2, Xr9g).
- Clarified **Patch Encoder/Decoder terminology** and cross-attention directions (Reviewers Xr9g).
- Added **Spotify dataset licensing details** (Reviewers Xr9g).

We hope these revisions address all concerns and further strengthen the paper.

Thank you again for your time and helpful feedback.

Best regards,
Authors

---

### Meta-Review · Area_Chair_yPyz · 2026-01-06

**Summary:**

The paper presents a novel curriculum schedule to learn alignment-aware speech patches that better aligns speech and text token granularity. The experiments are well designed to validate in both compute-controlled and data-controlled setups, the proposed LST improves on both S->S and T->T.
The reviewers all agree with the proposed method is of sufficient novelty and addresses an import problem in speech-text modeling.
The reviewers raised concerns on the experimental justification, which are addressed with additional results in the authors' rebuttal. Other clarification questions were also properly addressed.
Given the concerns are well addressed and LST is an extension from BLT tailed for speech-text with alignment information injected via curriculum schedule. I hence recommend accept.

**Reviewer Concerns:**

Reviewer fuu2:
* Add explanations on BLT which the proposed LST was inspired from: addressed
* evaluation on real human speech and common evals, for both understanding and TTS: addressed with additional ASR and TTS results.

SwjZ:
* unclear evaluation procedure, small test coverage, generation tasks: more explanation in rebuttal, added additional TTS results,

cWAg:
* impact on fine-grained acoustic-phonetic information, missing results on benchmarks like sWUGGY and sBLIMP: added in the response
* variability in single-run static experiments: addressed with averaged results over 3 independent runs.


Reviewer Xr9g:
* novelty: not ground breaking, but of sufficient novelty
* aligner choice, sufficiency, curriculum vs. sil merged, computation methodology: addressed in audio responses

**Reviewer Scores:**

Reviewer fuu2: rating 2, i would expect to be adjusted to 6/7.
Reviewer SwjZ: rating 6, no change
Reviewer cWAg: 10, no change
Reviewer Xr9g: 6, no change

---

### Decision · Program_Chairs · 2026-01-26

Accept (Oral)